# PAY ATTENTION TO WHAT MATTERS

## ABSTRACT

Despite the remarkable success of Large Language Models (LLMs), they still exhibit a limited capability to align their outputs to the user instructions. In this work, we introduce a simple and effective method, which we name GUIDE, that mechanistically increases attention scores in instruction tokens. To support this operation, we present *Influence*, a novel metric that highlights how the user's instructions propagate through the transformer layers and impact the LLM output. Our results show that GUIDE improves the accuracy of following instructions 29.4% to 60.4%, outperforming natural prompting alternatives and Supervised Fine-Tuning up to 1M tokens.

## 1 INTRODUCTION

Large Language Models (LLMs) are currently the state-of-the-art of most NLP tasks. Despite this success, pretrained LLMs sometimes struggle to accurately interpret diverse users' instructions and may generate outputs that do not align with human expectations. Additionally, LLMs may produce biased or hallucinated facts, which can limit their practical usefulness. Previous work (Kuratov et al., 2024; Lu et al., 2024b) indicate that transformers are less prone to align with instructions as the context length grows (Kuratov et al. (2024); Lu et al. (2024b)). In such cases, rather than fulfilling the user's request, the model generates nonsensical text or repeat segments from the prompt.

A common solution to this problem is Supervised Fine-Tuning (SFT) and Reinforcement Learning (RL). However, these approaches are resource-intensive, time-consuming, and sensitive to the specific data and task. Ideally, a more efficient approach would be one that, once implemented, does not require additional training.

In that sense, due to its low cost and broad accessibility, prompt engineering is widely used to align the outputs of LLMs with user preferences. However, this method does not always produce consistent results and can be very unstable, as demonstrated in (Sclar et al., 2024).

In this work, we introduce GUIDE (**G**uided **U**nderstanding with **I**nstruction-**D**riven **E**nhancements), a novel and systematic approach that allows users to emphasize critical instructions in their prompts. GUIDE enables users to influence the attention given to specific tokens by simply enclosing important text within tags like `<!->  <-!>` (as shown on Figure 1(a)). These special tags directs the LLM's focus, which is done by adding a bias to the attention scores toward the tokens they enclose. Our implementation is open-source and designed for seamless integration. Our experiments demonstrate that GUIDE significantly increases the likelihood of the model following key instructions and retrieving crucial information designated by the user, outperforming natural prompting techniques.

While GUIDE does not require additional training, it does necessitate the careful selection of how much to increase attention weights. In our study, we propose default values for certain tasks, but we also recognize the need to quantify these adjustments. To address this, we introduce a novel metric called *Influence*. This metric measures the importance of specific tokens in relation to instruction tokens within the text, and we use it to determine reasonable values for the increase in attention weights. We demonstrate that this metric correlates with the model's probability of following specific instructions.

To that end, the main contributions of this work are:

1. The introduction of GUIDE: a mechanistic approach for emphasizing instruction tokens, without need of any further computational resources.

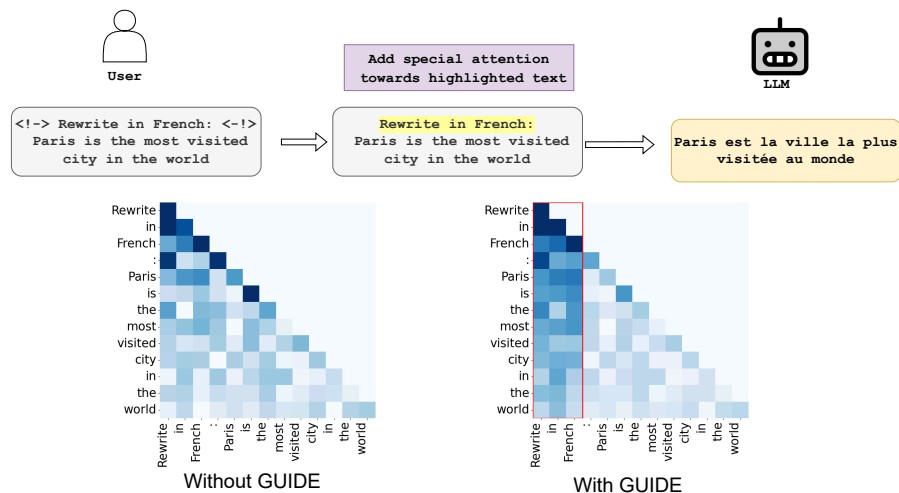

((a)) GUIDE uses tags (such as `<!->` `<-!>`) to know where to focus. It then enhances the importance of highlighted tokens by biasing the attention scores toward them, as shown by the attention matrices above, where each entry represents the impact of a past token (x-axis) on the ongoing token (y-axis).

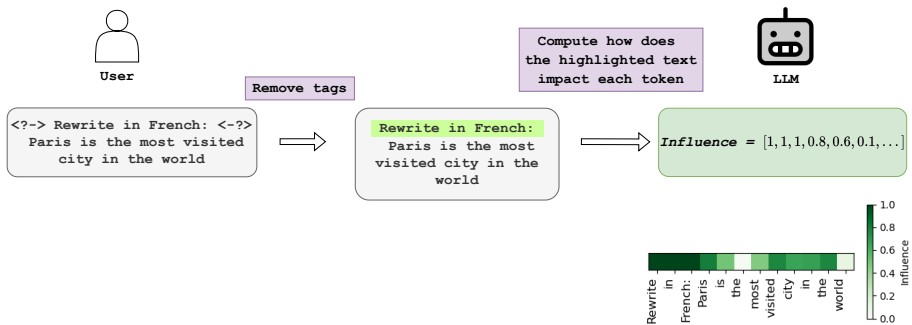

((b)) *Influence* is a metric that represents the impact of a sequence of tokens through context length. In our pipeline, it can be computed by enclosing the instruction within the tag `<?->` `<-?>`.

Figure 1: Schema of `PayAttentionPipeline`.

2. The introduction of *Influence*: a non-gradient metric that quantifies the importance of a given instruction over the text, which we use to adjust and understand GUIDE.

3. Release of `PayAttentionPipeline`: a HuggingFace-based implementation capable of performing generation with GUIDE and computing Influence (as illustrated in Figure 1).

## 2 RELATED WORK

**Alignment and instruction following**   Alignment techniques have the objective to *align* LLM outputs with human preferences. Model fine-tuning usually aligns the output of the LLMs with human intents using Reinforcement Learning with Human Feedback (RLHF) (Ouyang et al., 2022), Reinforcement Learning with AI Feedback (RLAIF) (Lee et al., 2023) or Direct Preference Optimization (DPO) (Rafailov et al., 2024). However, these methods have three significant constraints: they require specialized datasets, often with human annotations, thus reducing efficiency; they involve substantial computational complexity and cost due to the need for additional training; and they demand specialized expertise, as successfully implementing the process can be challenging. Given these constraints, this type of fine-tuning is typically reserved for general-purpose alignment, ensuring that models are Helpful, Harmless, and Honest (Shen et al., 2023) and are not suited to address end users' specific needs. Supervised fine-tuning (SFT) with techniques such as low-rank adapters (LoRA (Hu et al., 2021b)) offers a more accessible way to customize a model for individual user

requirements. However, these techniques still face the same three limitations, albeit to a lesser extent. Consequently, SFT is typically utilized only for very targeted use cases if used at all.

Utilizing LLMs for automated prompt engineering has demonstrated notable performance. Black-Box Prompt Optimization (BPO) is a sophisticated framework that automatically refines human-written prompts, often unstructured or ambiguous (Cheng et al., 2024). Similarly, the PE2 framework (Ye et al., 2024) enhances prompt performance by refining human-written prompts through a comprehensive search process. Although PE2 avoids additional model training, it increases complexity, latency, and cost, limiting its scalability. Both BPO and PE2 are generally designed for broad enhancements in prompt writing. They are not tailored to meet individual users' specific intentions or needs.

Due to its low cost and large accessibility, prompt engineering is extensively used to align the output of the LLMs with user preferences. This is clearly demonstrated by popular LLM frameworks like the system in (Yang et al., 2024), which empowers LM agents to tackle software engineering tasks. This system emphasizes crucial instructions through uppercase text and exclamation marks, like "PLEASE DO NOT DO THAT!" or "THE EDIT COMMAND REQUIRES PROPER INDENTATION." Similarly, the AI Scientist (Lu et al., 2024a), a leading system for automated scientific discovery, uses strong directives such as "ABSOLUTELY DO NOT ADD IT AGAIN!!!" to steer the model's behavior. These examples, drawn from highly influential frameworks widely used, underscore the pressing need for end-users to signal what matters most to them in order to guide LLMs toward better alignment with their goals. Currently, users rely on prompt engineering and forceful language to achieve this alignment. However, this approach does not consistently deliver positive results as shown in (Sclar et al., 2024).

In contrast, our proposed method, GUIDE, offers a reliable and systematic approach that enables users to mechanistically highlight critical instructions within the prompt.

**Explainability for Transformers** This is a particularly active area of research with many promising research directions. In the context of this work, we primarily focus on methods that attempt to quantify the significance of one token (or set of tokens) of interest.

Gradient metrics such as *Relevance* and *GradCAM* (Chefer et al. (2021a); Chefer et al. (2021b); Selvaraju et al. (2019)) have demonstrated promising results in Computer Vision, NLP and Text-To-Image tasks. Nonetheless, calculating gradients in large models, particularly those exceeding 7B parameters, demands substantial computational resources.

*Activation Patching* approaches (Meng et al. (2023); Zhang and Nanda (2024)) focuses on perturbing the inputs and checking how impactful will this perturbation be, based on where and how is it applied. They propose two types of perturbation: adding a gaussian noise in token embeddings, or perturbing the phrase in a semantical approach (changing important words in the phrase), and they evaluate the importance of each token by the difference of distributions of logits into these with perturbations.

Abnar and Zuidema (2020) introduced *Attention Rollout*, a method that uses attention weights to measure the impact of one token on another. This technique propagates attention weights through the layers to determine the importance of each token from any layer to the first layer. However, this method is primarily applicable to encoder-only architectures like BERT. In contrast, generative models typically use decoder-only architectures, making attention rollout less effectives.

In this work, we introduce Influence, a simple and computationally efficient metric specifically designed for decoder-only models.

## 3 GUIDE: (GUIDED UNDERSTANDING WITH INSTRUCTION-DRIVEN ENHANCEMENTS)

In this section, we present GUIDE, a novel and systematic approach that enables users to highlight critical instructions within the text input provided to an LLM. To understand how GUIDE operates, it is essential first to revisit the core mechanism of self-attention, which drives the functioning of LLMs.

## 3.1 DESCRIPTION OF THE METHOD

Each token $k$ in the input text is initially represented by an embedding, denoted by $E_k^{(0)}$, which undergoes progressive refinement through stacked layers of attention. By the time it reaches the final layer, $L$, this embedding $E_k^{(L)}$ is expected to encapsulate all the semantic information required to predict the next token, $k + 1$.

The process operates as follows: at each attention layer $\ell$, the embedding of a token $k$ is enhanced with the semantic information of past tokens ($i = 1, 2, \ldots, k - 1$) and itself. This enrichment occurs through a residual connection, where the embedding $E_k^{(\ell)}$ is updated with the output of the attention layer, which consists of a weighted average of the values $V_i^{(\ell)}$ of the past tokens. The vectors $V$, known as values, are derived from a simple linear transformation of the embeddings $E_i^{(\ell)}$, for $i \leq k$, and are responsible for carrying the semantic information from the past tokens.

The extent to which previous tokens influence the semantic update of the token $k$ is determined by attention logits, denoted by $w_{k,i}^{(\ell)}$. These logits represent the raw, unnormalized relevance scores between a token of interest $k$, called a key, and each preceding token $i \leq k$, called queries. The logits are then passed through a softmax function, which normalizes them to sum to one. The resulting normalized weights are known as attention scores ($\mathbf{A}^{(\ell)}$) and quantify the degree of influence each past token has on the current token's semantic representation at a given layer. Denoting $U_k^{(\ell+1)} := \text{Attention}^{(\ell+1)}(\mathbf{E}_k^{(\ell)})$, the operations at layer $\ell$ can be summarized as follows:[1]

$$E_k^{(\ell+1)} = E_k^{(\ell)} + U_k^{(\ell+1)} = E_k^{(\ell)} + \sum_{i=1}^{k} \mathbf{A}_{k,i}^{(\ell+1)} V_i^{(\ell)}, \tag{1}$$

The logits, and hence the attention scores, are automatically computed by the model. We argue that the end user should be able to influence the level of attention each token receives by explicitly signaling which instructions or pieces of information are critical. By doing so, the user can effectively guide the model to better align with his/her intention. We propose to achieve this by simply adding a bias, denoted by $\Delta$, to the attention logits of the important tokens, i.e., $\bar{w}_{k,i}^{(\ell)} = w_{k,i}^{(\ell)} + \Delta$, for all tokens $i$ indicated by the user. While this approach is direct, it proves to be highly effective, as demonstrated in the experimental results section.

## 3.2 CALIBRATING GUIDE

Using GUIDE, the addition of $\Delta$ directly increases the attention the model pays to the tokens of interest, amplifying their influence on the generated output. However, because attention scores must sum to one, this adjustment reduces the attention given to other tokens. If $\Delta$ is set too high, the model might overly focus on the highlighted tokens, which could disrupt the generation process. Therefore, it is crucial to select an appropriate $\Delta$ that balances these effects.

Our experiments suggest that for the Mistral and Gemma-2 models, a $\Delta$ of 2 works well for emphasizing instructions, while a $\Delta$ of 1 is effective for highlighting specific information within the text. Besides, using $\Delta$ values greater than 5 often led to nonsensical outputs (see Appendices D and H). Although these default values improve performance compared to natural prompting alternatives, the optimal choice of $\Delta$ depends on various factors, including the model, the nature of the task, etc. The most precise way to determine an appropriate $\Delta$ is through hyperparameter tuning on a validation set.

In this work, we also introduce a heuristic approach for calibrating $\Delta$ with just a couple of forward passes. The idea is to match the influence increase from $\Delta$ to a "natural" level that could be achieved through conventional prompting, such as using uppercase (see Figure 2). This calibration requires a metric that evaluates the influence of the selected tokens and tracks how this impact propagates both vertically across the stacked layers and horizontally across successive tokens.

---

[1]For simplicity, we have excluded the normalization and feedforward layers from this explanation.

## 3.3 INFLUENCE

Let us denote by $\bar{\mathcal{U}} = (x_1, \ldots, x_n)$ the overall sequence of tokens associated with the user's query and with $\mathcal{U} = (x_i, \ldots, x_j)$ the tokens related to the instruction that the user desires to highlight.

To maintain simplicity and minimize computational cost, we avoid using gradient-based metrics to evaluate the impact of a subset of tokens on the overall sequence (for example, see (Chefer et al., 2021a),(Chefer et al., 2021b), and (Selvaraju et al., 2019)). Instead, a more appropriate option appears to be the *Attention Rollout* method proposed in (Abnar and Zuidema, 2020). This metric can be easily computed during the forward pass, aligning well with our needs.

The *Attention Rollout* approach is based on a natural interpretation of attention scores. It postulates that the influence of a past token $i$ on the update of the current token $k$ is quantified by the attention score $\mathbf{A}_{k,i}^{(\ell)}$. The method addresses the residual connection by assuming that in the updated embedding $E_k^{(\ell+1)}$, both the previous embedding $E_k^{(\ell)}$ and the update vector $U_k^{(\ell+1)}$ contribute equally, each having an impact of $\frac{1}{2}$. The vertical and horizontal flow of the impact $\mathrm{R}_{\mathcal{U}}(E_k^{(l)})$ of a given token of interest $\mathcal{U}$ on an embedding $E_k^{(l)}$ is hence characterized by the following recurrence:

$$\mathrm{R}_{\mathcal{U}}(E_k^{(\ell)}) = \frac{1}{2}\left[\mathrm{R}_{\mathcal{U}}(E_k^{(\ell-1)}) + \mathrm{R}_{\mathcal{U}}(U_k^{(\ell)})\right] = \frac{1}{2}\left[\mathrm{R}_{\mathcal{U}}(E_k^{(\ell-1)}) + \sum_{i=1}^{k} \mathbf{A}_{k,i}^{(\ell)} \cdot \mathrm{R}_{\mathcal{U}}(E_i^{(\ell-1)})\right]$$

We argue that *Attention Rollout* inaccurately represents the flow of attention, particularly when handling the residual connection. The norm of the past embedding $E_k^{(\ell)}$ is typically about 100 times larger than that of the update vector $U_k^{(\ell+1)}$ (see Appendix B Figure 6(a)). .

As context length increases, one would expect that the importance of a subsequence of tokens would decrease, since the model must process more information. However, by assuming equal contributions from $E_k^{(\ell)}$ and $U_k^{(\ell+1)}$, *Attention Rollout* significantly overestimates the importance of past tokens.

This error compounds as the context length increases, leading to an inflated impact estimate that increases with the context and hence negatively correlates with the model's likelihood of following the token of interest, such as adhering to a specific instruction (Appendix B Figure 6(b)).

To address this issue, we introduce *Influence*, a new metric designed to quantify the impact flow of a token or a set of tokens of interest $\mathcal{U}$. This metric corrects Attention Rollout by weighting the contributions according to the norm of the vectors:

$$\mathrm{I}_{\mathcal{U}}(E_k^{(\ell+1)}) = \frac{\left(\|E_k^{(\ell)}\| \cdot \mathrm{I}_{\mathcal{U}}(E_k^{(\ell)}) + \|U_k^{(\ell+1)}\| \cdot \mathrm{I}_{\mathcal{U}}(U_k^{(\ell+1)})\right)}{\|E_k^{(\ell)}\| + \|U_k^{(\ell+1)}\|} \tag{2}$$

More precisely, *Influence* $\mathrm{I}_{\mathcal{U}} : \mathbb{R}^{dH} \to \mathbb{R}^+$ (where $d$ is the attention head dimension and $H$ is the number of attention heads), is a transformer interpretability metric designed to quantify how sequences of tokens in the user's query impact each others and relate to the LLM output. It is designed based on the following principles:

**Initialization:** We initialize the Influence value as 1 for tokens within the instruction $\mathcal{U}$, and as 0 elsewhere. Let $E_k^0 \in \mathbb{R}^{dH}$ be the embedding of token $x_k$. Then, the Influence initialization can be formally defined as:

$$\mathrm{I}_{\mathcal{U}}(E_k^0) = \mathbb{1}_{\{x_k \in \mathcal{U}\}}. \tag{3}$$

**Propagation Rules:** Given $m$ embedding vectors $E_1, \ldots, E_m \in \mathbb{R}^{dH}$, the joint Influence of the instruction tokens $\mathrm{I}_{\mathcal{U}} : \mathbb{R}^{dH} \times \cdots \times \mathbb{R}^{dH} \to \mathbb{R}_+$ is calculated as the average of each individual Influence, weighted by the norms of each embedding, as follows:

$$\mathrm{I}_{\mathcal{U}}(E_1, E_2, \ldots, E_m) = \frac{\sum_{i=1}^{m} \mathrm{I}_{\mathcal{U}}(E_i)\|E_i\|}{\sum_{i=1}^{m} \|E_i\|}. \tag{4}$$

Additionally, we maintain the invariance of Influence to function composition, i.e.,

$$\mathrm{I}_{\mathcal{U}}(f(E)) = \mathrm{I}_{\mathcal{U}}(E). \tag{5}$$

See Appendix A for detailed derivations.

Using *Influence*, we can calibrate GUIDE by choosing a $\Delta$ that mimics natural attention enhancement, such as writing in uppercase (Figure 2). This can be easily achieved with two forward passes, one with uppercase text and one without. $\Delta$ is then defined as the difference in log-influence between the two versions. It is important to note that our experiments indicate that an instruction highlighted with GUIDE typically has a greater impact on text generation compared to one highlighted with natural prompting, such as using uppercase, even if their influence scores are similar.

Moreover, in the absence of GUIDE, our experiments demonstrate that Influence correlates positively with the likelihood of a set of tokens impacting the model's output, such as following an instruction (e.g., summarizing in French) or retrieving specific information (e.g., finding a needle in a haystack). Thus, although not flawless, *Influence* offers a tool of independent value that can be used to compare and predict the impact of different natural prompting techniques.

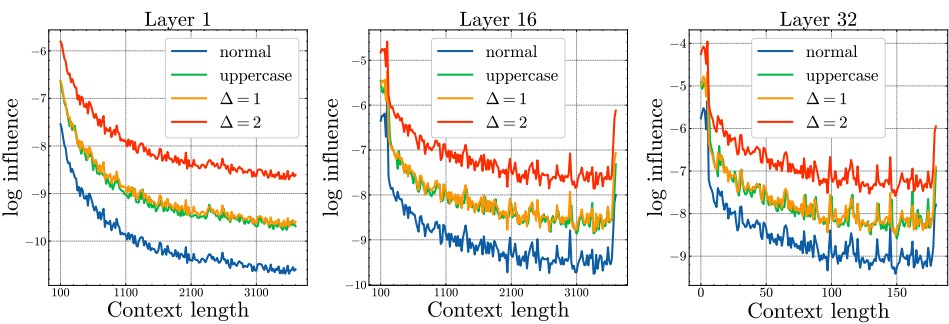

Figure 2: Log of the influence across different layers. This illustrates that with an appropriately chosen $\Delta$, GUIDE can effectively replicate—and even further amplify—semantically intuitive instructions, like using uppercase text.

## 4 EXPERIMENTS

In this section, we evaluate the benefits of GUIDE and Influence using Mistral-7b Instruct (Jiang et al., 2023) for following key instructions and retrieving crucial information designated by the user. In Appendix F, we conduct the same experiments with Gemma2-2b Instruct Team et al. (2024) with the same values of $\Delta$, and we obtain very similar results.

### 4.1 DESCRIPTION

**Summarization in French**  To evaluate the capability of GUIDE to support LLMs in producing outputs aligned with the user's query, we perform experiments related to text translation and summarization. In these experiments, we have used text from OpenWebText (Gokaslan and Cohen, 2019), chosen for its variety in context lengths. We have divided the dataset into groups based on context length, containing texts from a 500-token window, such as (0, 500], (500, 1000], and so on. From each group, we randomly selected 20 texts and generated 10 summaries for each text using multinomial sampling (Wiher et al., 2022).

**A needle in a haystack**  To evaluate the impact of our approach on the model's ability to retain information, we have conducted the Needle in a Haystack. This test involves embedding specific information at a particular location within a text and then asking a question related to that information at the end of the text. Our hypothesis is that by adding extra attention to this text, the model's outputs would improve, as the final representation should be more closely aligned with the information tokens. We have followed the methodology outlined by (Kamradt, 2023). Specifically, we have inserted specific information, referred to as the "needle" at variable positions within a given text. After this insertion, we have asked a question to the LLM related to the inserted information (see the complete prompt in Appendix G).

To conduct this experiment, we have sampled 200 texts from the OpenWebText (Gokaslan and Cohen, 2019) dataset, selecting 50 texts for each context window of size 500, ranging from 0 to 6000 tokens. For each text, the needle was inserted at 10 different quantiles (10%, 20%, ..., 100%). We placed the needle immediately after a period ('.') to maintain the semantic integrity of the text.

**JSON Generation**   To assess the efficiency of GUIDE in generating outputs in a specified format, we have conducted experiments focused on JSON generation. For our inputs, we have used texts from books written between 1510 and 1699, sourced from the BL Books dataset (Labs, 2021). We have prompted the model to extract and generate key information about each book in a predetermined JSON format, as detailed in G. We have randomly selected 300 books from the BL Books dataset and divided each text into context length windows of 500 tokens, ranging from 0 to 4000 tokens. These text segments were then incorporated into our template, where the Mistral model was expected to generate a JSON output that precisely followed the specified format.

We have inputted special attention into the tokens of `Your response should follow exactly this template` and we have then evaluated the Jaccard index between the keys of the generated JSON and the schema.

**Influence**   For each of the experiments mentioned above, we have evaluated the relationship between the *Influence* metric and the probability of obtaining correct outputs and compare it with *Attention Rollout* and raw attention scores. We observe that, attention rollout shows a negative correlation and an AUC below $0.5$ in two out of three experiments, supporting our hypothesis that attention rollout may not accurately reflect the model focus. Also as expected, we see that raw attention has a random behavior in two of three setups, with ROC AUC scores around $0.5$. *Influence* shows a strong positive correlation and ROC AUC with the likelihood of following instructions, supporting our hypothesis that it better quantifies attention flow compared to other non-gradient metrics. Due to space limitations, further details are left for the appendix (Appendix C).

### 4.2 RESULTS

**Summarization in French**   We have conducted experiments with GUIDE, biasing attention scores towards the instruction `Summarize in French`. Fig. 3(a) shows the observed probability that the LLM summary is in French when using GUIDE and compares the results achieved with the baseline model, with both uppercase and normal prompts, as well as the performance observed when including 'Important:' before the prompt instruction. Our findings show that GUIDE leads to an improvement from $29.4\%$ to $60.4\%$ with respect to the raw model, and that the best result is achieved with $\Delta = 2$. Besides, to confirm that GUIDE does not induce a deterioration of the quality of the generated outputs, we compare the summaries generated in French obtained with the raw prompt and the ones obtained with GUIDE. We observed no noticeable degradation. Further details can be found in Appendix D.

As a baseline, we compare the performance of GUIDE to prompt engineering and Supervised Fine-Tuning (SFT) using LORA (the hyperparameters can be found in Appendix E). Figure 3(a) show that using uppercase or adding 'Important' on the instruction does not provides notable improvements, consistently underperforming GUIDE, while Figure 3(b) shows that GUIDE outperforms SFT until 1M training tokens. These results confirms that our method is an effective solution for aligning LLMs to instruction following that does not require additional training.

**Needle in a haystack**   Figure 4 shows the probability of outputting the correct phrase over the context length and the position of the needle, respectively. The Mistral model demonstrates stable performance across varying context lengths and needle positions within this window. As expected, the addition of $\Delta$ to the needle tokens consistently enhances performance from $87.0\%$ to $92.1\%$, with optimal values of $\Delta$ around 1. We can also note that, on average, the LLM is more effective at retrieving information when it is located at the beginning or the end of the text. This is in accordance with previous results (Kuratov et al., 2024; Kamradt, 2023).

**JSON Generation**   We measure the Jaccard index between the keys of the generated JSON and the keys on the schema. We observed that the optimal value for $\Delta$ is approximately 3, resulting in an average score improvement of 30% compared to the raw model (Figure 5). We also note that

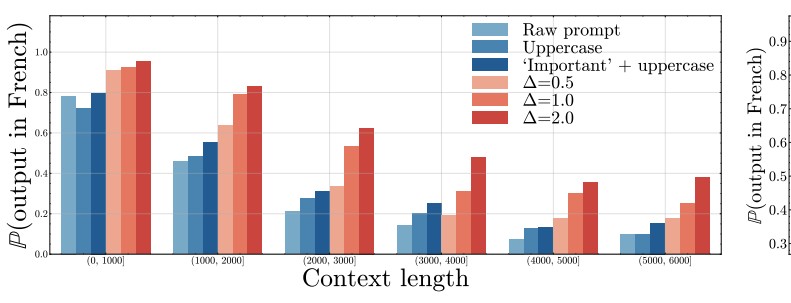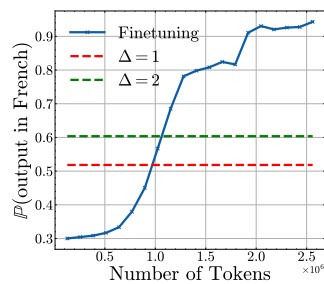

((a)) Probability of outputting a summary in French.

((b)) Performance of SFT over number of tokens (in millions) used during training.

Figure 3: Summarization results: (a) GUIDE outperforms prompt engineering techniques like using uppercase text, and (b) GUIDE demonstrates greater accuracy than SFT up to 1 million training tokens.

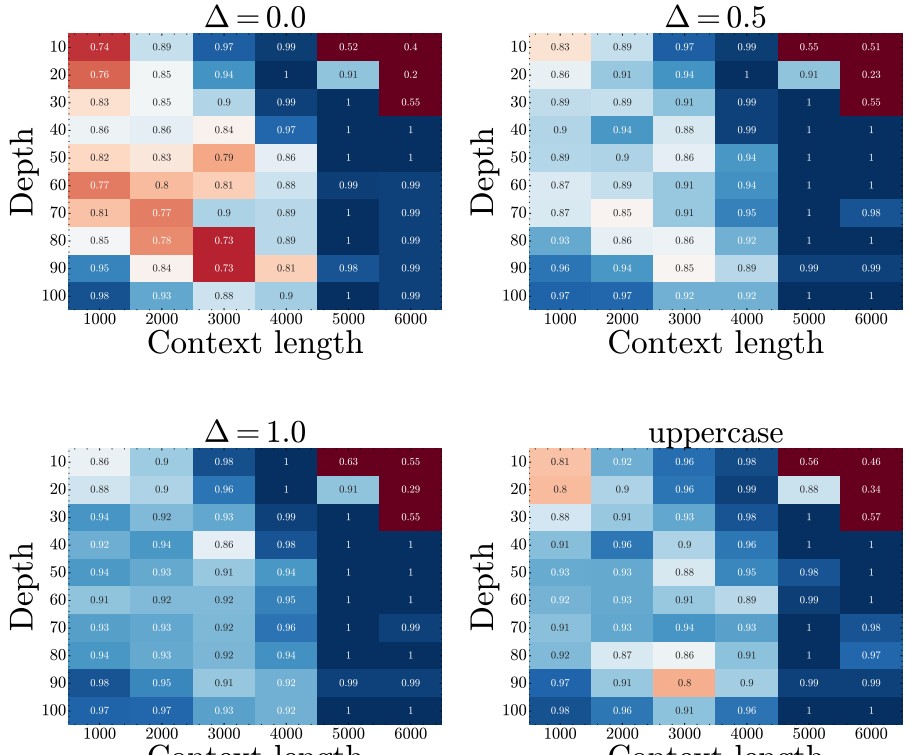

Figure 4: Heatmap of scores in a needle in haystack test.

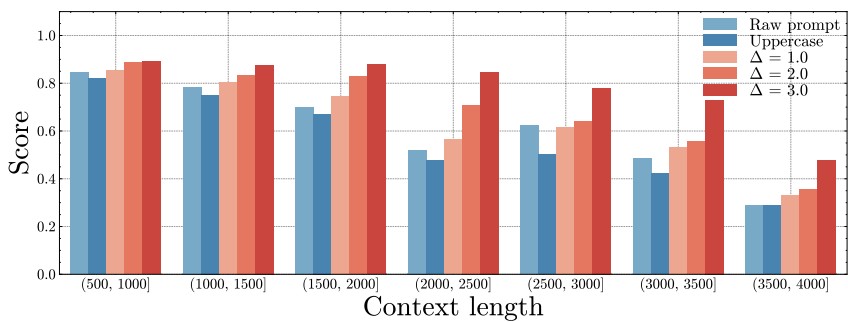

Figure 5: Jaccard Index vs Context length for the JSON generation experiment.

in almost every generation the scores were $0$ or $100\%$. This indicates that most of the time, the generated output was either a perfect match to the requested schema or not in JSON format at all.

## 5 CONCLUSION

While Transformers represent the state-of-the-art in almost all NLP tasks, they often exhibit unexpected behaviors, particularly hallucination, which becomes more pronounced as context length increases. This work introduces GUIDE, a mechanical approach for instruction alignment that does not require further optimization. We demonstrate that GUIDE effectively mitigates hallucination in instruction-following scenarios without significantly compromising output quality. We also introduce Influence, a novel metric for Transformer explainability that quantifies the importance of subsequences of tokens within the context and can be used to calibrate GUIDE at low cost. Both GUIDE and Influence are made publicly available in a Hugging Face-based pipeline.

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

## A  DETAILED DERIVATIONS ON INFLUENCE

Let us denoted by $d$ the transformer head dimension, with $H$ the number of attention heads, and with $s$ the context length. Following the propagation of a transformer layer (Vaswani et al., 2023), the embedding on layer $\ell$. $\mathbf{E}^{(\ell)}$ is computed as follows:

$$\mathbf{E}^{(\ell)} = \text{Linear}\left(\text{Norm}\left(\mathbf{E}^{(\ell-1)} + \text{Attention}^{(\ell)}(\mathbf{E}^{(\ell-1)})\right)\right), \tag{6}$$

$$\text{Attention}^{(\ell)}(\mathbf{E}) = \mathbf{A}^{(\ell)} \cdot V^{(\ell)}(\mathbf{E}), \tag{7}$$

where, $\mathbf{E}^{(\ell-1)} \in \mathbb{R}^{dH \times s}$ is the embedding on layer $\ell - 1$, $\mathbf{A}^{(\ell)}$ is the attention matrix on layer $\ell$, $V^{(\ell)} : \mathbb{R}^{dH} \to \mathbb{R}^{dh}$ is a linear function that maps the token embeddings to the values vector, Norm is a normalization function, and Linear is a conventional multilayer perceptron (MLP) function. Then, we can compute the Influence of token $k$, $\mathbf{E}_k^{(\ell)}$, as follows:

$$\begin{aligned}
I_{\mathcal{U}}(E_k^{(\ell)}) &= I_{\mathcal{U}}\left(\text{MLP}\left(\text{Norm}(\mathbf{E}^{(\ell-1)} + \text{Attention}^{(\ell)}(\mathbf{E}^{(\ell-1)}))_k\right)\right) \\
&= I_{\mathcal{U}}\left((\mathbf{E}^{(\ell-1)} + \text{Attention}^{(\ell)}(\mathbf{E}^{(\ell-1)}))_k\right) \\
&= \frac{I_{\mathcal{U}}(E_k^{(\ell-1)}) \cdot \|E_k^{(\ell-1)}\| + I_{\mathcal{U}}(\text{Attention}^{(\ell)}(\mathbf{E}^{(\ell-1)})_k) \cdot \|\text{Attention}^{(\ell)}(\mathbf{E}^{(\ell-1)})_k\|}{\|E_k^{(\ell-1)}\| + \|\text{Attention}^{(\ell)}(\mathbf{E}^{(\ell-1)})_k\|} \\
&= \frac{I_{\mathcal{U}}(E_k^{(\ell-1)}) \, r_k^{(\ell-1)} + I_{\mathcal{U}}(\text{Attention}^{(\ell)}(\mathbf{E}^{(\ell-1)})_k)}{1 + r_k^{(\ell-1)}}
\end{aligned} \tag{8}$$

where $r_k^{(\ell-1)} := \frac{E_k^{(\ell-1)}}{\text{Attention}^{(\ell)}(\mathbf{E}^{(\ell-1)})_k}$.

Influence is computed recursively over layers, *i.e.*, when we compute the Influence on layer $\ell$, we have already computed the Influence on layers $1, \ldots, \ell - 1$. This means that $I_{\mathcal{U}}(E_k^{(\ell-1)})$ is already computed, while we still need to compute $I_{\mathcal{U}}(\text{Attention}^{(\ell)}(\mathbf{E}^{(\ell-1)})_k)$. Developing equation 7:

$$\text{Attention}^{(\ell)}(\mathbf{E}^{(\ell-1)})_k = \sum_{i=1}^{s} \mathbf{A}_{k,i}^{(\ell-1)} E_i^{(\ell-1)},$$

$$I_{\mathcal{U}}(\text{Attention}^{(\ell)}(E^{(\ell-1)})_k) = \frac{\sum_{i=1}^{s} \mathbf{A}_{k,i}^{(\ell-1)} E_i^{(\ell-1)} \, I_{\mathcal{U}}(E_i^{(\ell-1)})}{\sum_{i=1}^{s} \mathbf{A}_{k,i}^{(\ell-1)} E_i^{(\ell-1)}}.$$

Then, if we approximate the norm of the embeddings $E_i^{(\ell-1)}$ with a constant, we obtain a simplified expression

$$I_{\mathcal{U}}(\text{Attention}^{(\ell)}(E^{(\ell-1)})_k) = \sum_{i=1}^{s} \mathbf{A}_{k,i}^{(\ell-1)} I_{\mathcal{U}}(E_i^{(\ell-1)}).$$

With this approximation, equation 8 becomes

$$I_{\mathcal{U}}(E_k^{(\ell)}) = \frac{I_{\mathcal{U}}(E_k^{(\ell-1)})}{1 + r_k^{(\ell-1)}} r_k^{(\ell-1)} + \frac{\sum_{i=1}^{s} \mathbf{A}_{k,i}^{(\ell-1)} \, I_{\mathcal{U}}(E_i^{(\ell-1)})}{1 + r_k^{(\ell-1)}}. \tag{9}$$

## B    ILLUSTRATION OF THE LIMITATION OF *Attention Rollout*

We argue that *Attention Rollout* inaccurately represents the flow of attention, particularly when handling the residual connection. The norm of the past embedding $E_k^{(\ell)}$ is typically about 100 times larger than that of the update vector $U_k^{(\ell+1)}$ (see Appendix B Figure 6(a)). .

As context length increases, one would expect that the importance of a subsequence of tokens would decrease, since the model must process more information. However, by assuming equal contributions from $E_k^{(\ell)}$ and $U_k^{(\ell+1)}$, *Attention Rollout* significantly overestimates the importance of past tokens.

This error compounds as the context length increases, leading to an inflated impact estimate that increases with the context and hence negatively correlates with the model's likelihood of following the token of interest, such as adhering to a specific instruction (Appendix B Figure 6(b)).

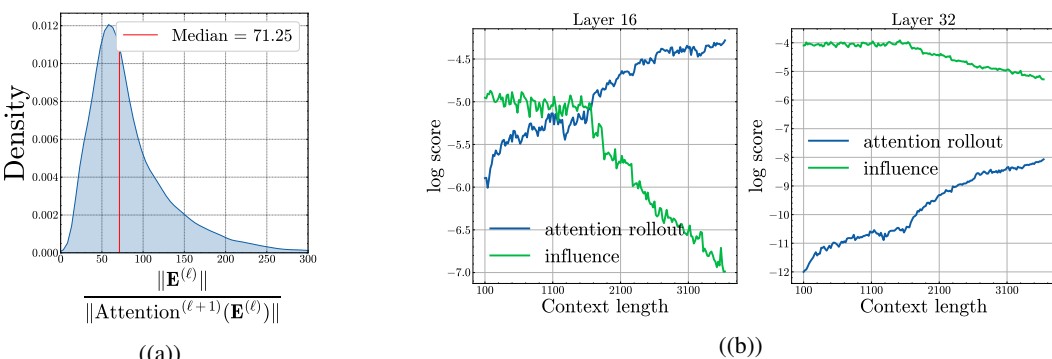

((a))                                                                ((b))

Figure 6: (a) : Distribution of ratio between norms of token embeddings before and after attention; (b): Attention rollout ($R_{\mathcal{U}}(E_k^{(\ell)})$) and Influence ($I_{\mathcal{U}}(E_k^{(\ell)})$) trends in log scale over context length ($k$) in intermediate and final layers ($\ell = 16$ and $\ell = 32$). The instruction tokens $\mathcal{U}$ were situated on the beginning of the prompt.

## C    EVALUATION OF INFLUENCE

For each of the experiments mentioned in the main text, we have evaluated the relationship between the *Influence* metric and the probability of obtaining correct outputs. To achieve this, we have calculated both the ROC AUC score and the correlation between the importance of instruction tokens and the last token in the sequence. Then, we have compared these results through non-gradient metrics, such as Attention Rollout and raw attention scores. Our hypothesis is that Influence has a strong positive correlation with the probability of correct outputs.

Given that the ROC AUC is a classification metric, it was necessary to binarize our scores. In the French summarization experiment, we have done this by assigning a score of 1 to texts in French and 0 to those in other languages. In the "needle in a haystack" experiment, a score of 1 was given to prompts that successfully identified the needle information, while those that did not were assigned a 0. Similarly, for the JSON generation experiment, outputs that adhered to the JSON format were assigned a 1, and those that did not were assigned a 0.

**Results**    Table 1 shows the correlation and ROC AUC of each metric to correct output. We note that, attention rollout shows a negative correlation and an AUC below $0.5$ in two out of three experiments. This observation supports our initial hypothesis that attention rollout may not accurately reflect the model focus. Also as expected, we also see that raw attention has a random behavior in two of three setups, with ROC AUC scores around $0.5$.

The stronger positive correlation and ROC AUC between *Influence* and the likelihood of following instructions supports our hypothesis that our metric better quantifies the attention flow in a Transformer than other existing non-gradient metrics.

Table 1: AUC and correlation of our metric and the probability of generating a correct output

|  | Metric | ROC AUC | Correlation |
|---|---|---|---|
| **Summarizing in french** | Influence | 0.74 | 0.72 |
|  | Attention rollout | 0.24 | -0.35 |
|  | Raw attention | 0.58 | 0.13 |
| **A needle in a haystack** | Influence | 0.62 | 0.12 |
|  | Attention rollout | 0.55 | 0.10 |
|  | Raw attention | 0.48 | -0.03 |
| **JSON generation** | Influence | 0.63 | 0.23 |
|  | Attention rollout | 0.31 | -0.29 |
|  | Raw attention | 0.64 | 0.23 |

## D    EVALUATION OF THE QUALITY OF OUTPUTS

In addition to verifying that the LLM summary is in French in Section 4.2, we have also evaluated the quality of the outputs using BERTScore (Zhang et al., 2020), calculated in comparison to target summaries generated by a Llama 3 70B model (AI@Meta, 2024).

To highlight the pertinence of BERTScore, in evaluating the quality of the summaries, we show in Fig. 7 the distribution of the observed BERTScore conditioned to the generated text being in French or not. We observe that the distribution for texts generated in French is shifted to the right compared to those not in French, indicating that BERTScore is a suitable metric for assessing the quality of generated texts.

To measure the impact of GUIDE on the quality of the LLM outputs, we have evaluated the winning rate by comparing the quality of the texts generated with and without GUIDE in terms of BERTScore. Specifically, for each pair of texts generated in French $(t_{i,\Delta}, t_{i,\text{raw}})$ by GUIDE and the unmodified (raw) model, we have determined which text had a higher BERTScore. Table 2 shows that for small enough choice of $\Delta$, the quality of the output is not highly affected, with winning rates of $50.5\%$ for $\Delta = 0.5$ and $\Delta = 1$ and $49\%$ for $\Delta = 2$. These results indicate that GUIDE maintains the model's capability to generate semantically correct text. However, as mentioned in Sec. 3.2, larger values of $\Delta$, e.g. $\Delta = 5$ results in poor outputs (see also Appendix H).

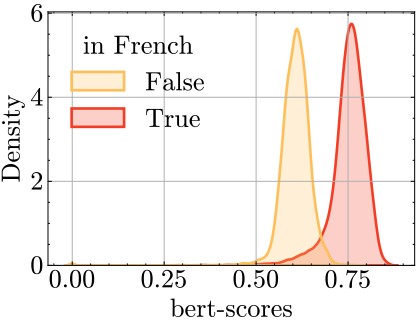

Figure 7: Distribution of Bert scores conditioned to the generated text being in French or not.

Table 2: Winning rate of text generated in French for GUIDE versus the baseline model (Mistral 7B).

| $\Delta$ | Winning Rate |
|---|---|
| 0.5 | 50.5% |
| 1 | 50.5% |
| 2 | 49% |
| 5 | 38.5% |

## E    SUPERVISED FINETUNING HYPERPARAMETERS

In our supervised fine-tuning experiments, we leveraged LoRa techniques Hu et al. (2021a), setting the sequence length to 8192 using sample packing and block-attention to prevent cross-sample contamination. We configured the LoRa rank to 64 and set the alpha parameter to 16. For regularization, we applied a dropout rate of 0.05. To maximize the adapter's expressiveness, our

LoRa implementation targeted all modules. The batch size was set to 128k tokens, with a maximum learning rate of 1e-4, following a cosine scheduler with a 10-step warm-up.

## F    EXPERIMENTAL RESULTS WITH OTHER MODELS

We have conducted the same studies presented in Section 4 also using the Gemma 2 - 2B Instruct model (Team et al., 2024). Our results indicate that, even with smaller models, GUIDE can still improve the accuracy of following instructions, increasing the accuracy from $43.4\%$ to $59.8\%$ on summarization in French , $65.2\%$ to $77.5\%$ on retrieval and $14.4\%$ to $24.1\%$ on JSON generation (see Tables 3, 4 and 5).

Table 3: Summarization in French results for Gemma 2- 2B

| $\Delta$ | Score |
|---|---|
| 0.0 | 0.434 |
| Uppercase | 0.408 |
| 0.5 | 0.514 |
| 1.0 | 0.571 |
| 2.0 | **0.598** |
| 5.0 | 0.002 |

Table 4: Needle in a Haystack results for Gemma 2- 2B

| $\Delta$ | Score |
|---|---|
| 0.0 | 0.652 |
| Uppercase | 0.766 |
| 0.5 | 0.722 |
| 1.0 | **0.775** |

Table 5: JSON Generation results for Gemma 2- 2B

| $\Delta$ | Score |
|---|---|
| 0.0 | 0.144 |
| Uppercase | 0.211 |
| 1.0 | 0.190 |
| 2.0 | 0.241 |
| 3.0 | **0.241** |

## G    PROMPTS USED IN EXPERIMENTS

---
**Summarization in French**

Summarize in French

{context}

---

---
**A needle in a haystack**

<question>
Your objective is to answer the following question based on the context:

{question}

Don't give information outside the document or repeat our findings
</question>

{context with needle}

<question>
Your objective is to answer the following question based on the context:

{question}

Don't give information outside the document or repeat our findings
</question>

---

---

**JSON generation**

You are an assistant designed to provide information in JSON format.
I will give you a story, and you need to extract and return specific details from the story.
Do not output anything else than the JSON.
Your response should follow exactly this template:

```
<schema>
{
    "title": "title of the story (string)",
    "genre": string,
    "characters":
      [
          {
              "name": string,
              "description": string. If not available set it to none
          }
      ] (one dict per character),
    "author": "the author of the story. If not available, set it to None",
    "summary": "a brief summary of the story. Do not write more than 50 words",
    "date": "when the story was released (string)",
    "scenery": "where the story takes place (string)",
}

</schema>

{content}
```

---

## H   EXAMPLES OF POOR GENERATION OF TEXT

This Appendix provides examples of the model failing to follow user instructions.

### H.1   RAW GENERATION (WITHOUT GUIDE)

A common error occurs when the model simply repeats the given prompt. This example has been obtained using multinomial sampling at generation time and without GUIDE ($\Delta = 0$).

---

**Prompt**

Summarize in French:
The red Ford Expedition mounted the sidewalk along DeKalb Avenue in Fort Greene, Brooklyn, on Saturday afternoon not once, but twice.
The first time, it narrowly missed two people at the southwest corner of Clermont Avenue, crashing into a parked car, jumping the curb and hitting another vehicle, the police said.
But the second time, pedestrians along the busy Brooklyn thoroughfare were not so lucky: The S.U.V. struck and killed a 9-year-old boy, Lucian Merryweather, who was with his mother on the northeast corner. His 5-year-old brother and a woman who had been in the crosswalk were hurt.
At first glance, the crash seemed likely to attract attention but unlikely to result in criminal charges. The driver, Anthony Byrd, 59, of Clinton Hill, remained at the scene and was not intoxicated, the police said.

---

> **Output**
>
> The red Ford Expedition mounted the sidewalk along DeKalb Avenue in Fort Greene, Brooklyn, on Saturday afternoon twice. The first time it narrowly missed two people and crashed into a parked car, jumping the curb and hitting another vehicle. However, during the second time, pedestrians were not so lucky as the S.U.V. struck and killed a 9-year-old boy, Lucian Merryweather, who was with his mother on the northeast corner. His 5-year-old brother and a woman who had been in the crosswalk were hurt. Initially, the crash appeared to be unlikely to attract attention and result in criminal charges, as the driver, Anthony Byrd, 59, of Clinton Hill, remained at the scene and was not intoxicated, the police said.

The LLM sometimes generates a very limited number of tokens that, although present in the prompt, are nonsensical when presented alone (and do not fulfil the user task). In this instance, we employ multinomial sampling to generate the text (without GUIDE).

> **Prompt**
>
> Summarize in French:
> Vine
> Oh, those crazy Sacramento Kings.
> On Saturday, head coach George Karl raised some eyebrows by saying guard Seth Curry would only last in the NBA for "a couple years," per Blake Ellington of Sactown Royalty.
> According to Ellington, Curry posted (and deleted) this tweet:
> Look familiar?
> A day later, the 25-year-old started and knocked down four of seven shots from the field en route to a 14-point performance in a 133-111 win over the Dallas Mavericks. During Curry's postgame interview, DeMarcus Cousins—rocking a sweet towel around his head—chimed in.
> "You gon' be in the league a long time, boy," the big fella said emphatically. "Remember that. A long time."
> Maybe it's a coincidence, and the All-Star was simply supporting a teammate who's been struggling.
> But Cousins and Karl's relationship has been volatile ever since the 64-year-old took the Kings job in 2014-15.
> Snake emoji, indeed.
> Update
> Regardless of the reason for Cousins' support, big brother Stephen Curry approved, reacting to the incident with a 100-laden tweet:
> [Vine]

> **Output**
>
> Tweet: Look familiar?

## H.2 GENERATION WITH GUIDE USING UNREASONABLY LARGE $\Delta$

We have noted that $\Delta > 5$ very often produces incorrect/nonsensical output. In the following example, we have set $\Delta = 9$ and performed greedy decoding.

**Prompt**

<!!-> Summarize in French <-!!>
Paris is the capital and largest city of France. With an official estimated population of 2,102,650 residents in January 2023 in an area of more than 105 km2 Paris is the fourth-largest city in the European Union and the 30th most densely populated city in the world in 2022. Since the 17th century, Paris has been one of the world's major centres of finance, diplomacy, commerce, culture, fashion, and gastronomy. For its leading role in the arts and sciences, as well as its early and extensive system of street lighting, in the 19th century, it became known as the City of Light

**Output**

1. 19th century, 19th century, 19th century, 19th century, 19th century, 19th century, 19th century, 19th century, 19th century, 19th century, 19th century, 19th century, 19th century, 19th century, 19th century, 19th century,

