# OpenReview forum: "Pay Attention to What Matters"
_ICLR.cc/2025/Conference — ICLR 2025 Conference Withdrawn Submission_

### Official Review · Reviewer_Bt72 · 2024-10-28

**Soundness:** 2
**Presentation:** 1
**Contribution:** 1
**Rating:** 3
**Confidence:** 4

**Summary:**

- This study introduces a prompt engineering method for large language models which increases attention scores in instruction tokens by simply enclosing important text within tags like <!-> <-!>.
- In addition, the study presents a metric that highlights how the user’s instructions propagate through the transformer layers and impact the LLM output.
- Experimental results show that the method improves the accuracy of following instructions.

**Strengths:**

- This study introduces a prompt engineering method for large language models which increases attention scores in instruction tokens by simply enclosing important text within tags like <!-> <-!>.
- A new metric is proposed to clarify that the proposed method focuses on the instruction text that are highlighted by the proposed method.

**Weaknesses:**

- There is already a prior research on prompt engineering method that shows that enclosing the tags improves performance, but this study does not describe the relationship. e.g. https://arxiv.org/abs/2309.13078
- Assessing the impact of attentions from the lens of vector norms is already a hot topic, but there is no description about the relationships.
https://arxiv.org/pdf/2004.10102
- Lack of relationships between sections, mathematical symbols, and descriptions of key points in each section makes the manuscript difficult to read. For example, In 3.1 DESCRIPTION OF THE METHOD, which is a preliminary setup and which is the proposed method? Another example is that in 3.3 INFLUENCE, R_U function and I_U function at the first layer are not defined so that we cannot understand what those functions mean when we are given only the recursive formula.

**Questions:**

- Is it possible to propose an algorithm to automatically improve the methodology using the proposed metrics? The intent of the question is related with a suggestion to improve the contribution of this study.
- As the title “PAY ATTENTION TO WHAT MATTERS” suggests, it is important to put the attention on the important parts, but in some cases, the instruction sentence is not the only important part. For example, in the case of sentence summarization, it is necessary to give strong attention to important parts of long sentences instead of instruction sentences. What is the relationship between that case and this study?
- I assume that the experiment was done with zero-shot prompting (but it was unclear because of the lack of description), what are the differences if we conduct experiments with few-shot in-context learning?

---

### Official Review · Reviewer_hWtS · 2024-11-02

**Soundness:** 2
**Presentation:** 2
**Contribution:** 2
**Rating:** 3
**Confidence:** 4

**Summary:**

The paper introduces GUIDE, a method for improving Large Language Models' ability to follow instructions by emphasizing important tokens in prompts, using special tags that bias attention scores. GUIDE requires no additional training, making it efficient and accessible. To support this, the authors also introduce the Influence metric, which quantifies the impact of instructions on the model's output. Experiments show GUIDE significantly enhances alignment with instructions across tasks, outperforming other methods like fine-tuning​.

**Strengths:**

- The paper is tackling an important problem in LLM. For instance, as the user prompt gets longer, we observe the LLM not following every part of the user instruction. A method that can enhance instruction following is a great contribution to the field.
- The paper is clear and generally well written.
- Influence score seems to give a good insight into evaluating the impact of tokens on the entire sequence.

**Weaknesses:**

- It is missing a very relevant work from ICLR 2024, "Tell Your Model Where to Attend: Post-hoc Attention Steering for LLMs" by Zhang et al.
- While it is good to know that the llm is now producing French more often, but this may be at the cost of degraded output accuracy. I suspect that if the attention weights are manually changed, the quality of the outputs may be affected. I think it is very important to measure this as well.
- I think Influence metric shouldn't be called "metric", because a low influence score doesn't necessarily mean the model is not following the instruction well. I think "probe" is a more appropriate term, which helps us to perform intrinsic analysis.

**Questions:**

- Could you please describe your work with respect to Zhang et al. (ICLR 2024) as mentioned in the weakness?
- Could you measure the change in output quality as you apply your method?

---

### Official Review · Reviewer_fsgv · 2024-11-04

**Soundness:** 2
**Presentation:** 3
**Contribution:** 3
**Rating:** 5
**Confidence:** 5

**Summary:**

This paper tackles the instruction following problem and proposes an attention manipulation method to force the model to pay attention to the user instructions or important information in the context. The method is very simple: simply add a bias to the attention logits (before normalization)  for the instructions/important information. The authors also developed an "influence" score to guide the setting of the bias. Inference is a score that describes how much one part of the text affects the later part of the text. The authors set the bias such that the influence matches that when setting the instructions to be upper case.

The experiments are mostly synthetic or toy to clearly demonstrate the instruction following ability: summarization in French (metric: p(output in French), needle in a haystack, and JSON generation (Jaccard index of the keys). The experiments demonstrated the effectiveness of the proposed method, improving over both baselines and uppercasing the instructions.

**Strengths:**

(1) The proposed method is very simple and intuitive. The experiments clearly showed that the methods improved "instruction following" at a coarse granularity. Instruction following is an important topic in modern language model development and the proposed method could have a significant impact and adoption in real applications.

(2) The proposed influence score is an intuitive way to guide the setting of the attention bias.

**Weaknesses:**

(1) Since this is a training-free method, it would be even better if the authors conducted the experiments on more models with different scales. Given that the experimented models are not very big (<7B) and strong (there are better models now like Llama-3), the lack of comprehensive experiments with different models may undermine the credibility of the experiments.

(2) The tasks the authors selected are rather synthetic. Moreover, the metrics are too simple: for example, the correctness of the French translation and the information extracted for JSON generation should also be evaluated. Even though the focus of the method is instruction following, the correctness of the output should also be measured to understand if the method hurts the performance. Also, the authors should make the evaluation part clearer: for example, I did not find how "whether output is French" is evaluated.

(3) There are several existing instruction-following benchmarks, such as IFEval (Zhou et al.). Even though the existing instruction following benchmarks are also quite synthetic (mostly focusing on writing tasks), the instructions are at least more complex and the evaluation is more rigorous.

Zhou et al. Instruction-Following Evaluation for Large Language Models

**Questions:**

Please see "weakness".

Minor comment : the citation format seems to be wrong for in-place citation. You should use `\citet` for those and it should look like "... appears to be the attention rollout method proposed in Abnar and Zuidema (2020)."

---

### Official Review · Reviewer_Hoj7 · 2024-11-11

**Soundness:** 2
**Presentation:** 3
**Contribution:** 2
**Rating:** 3
**Confidence:** 4

**Summary:**

This paper presents 1) an inference-time attention enhancement mechanism to improve LLM's instruction-following capability given user's annotation (which part of the input is crucial instruction) 2) *Influence*: a simple yet reasonable improvement to *Attention Rollout* that considers the ratio between the residual stream and the attention output.

The proposed pipeline is constructed using 1) and 2). Specifically, for each input with marked instruction, the *influence* scores when the marked instruction tokens is capitalized is calculated, and a delta weight is added to the QK attention weights before SoftMax at the marked instruction tokens to mimic the same *influence* score to boost the importance of the tokens in the context.

The efficacy of the proposed pipeline is evaluated over three settings: French summarization, needle in a Haystack, and JSON generation. All results indicate that a well-calibrated attention weight augmentation can improvement performance of the LLM on the target task, outperforming the prompting baselines.

**Strengths:**

1) The presentation of the methods are fairly straightforward and clean.
2) The proposed method is very intuitive and simple.
3) Without any extra resources, a substantial improvement has been seen over all the selected datasets, making it potentially an appealing plug-and-play mechanism to be added to deployed model to highlight crucial part of the input.

**Weaknesses:**

Sec 3:
1) The concept of 'influence' as an enhancement to attention rollout is intriguing and initially seems valuable. However, the exact role of influence within the implementation remains somewhat ambiguous. At first the glance, the narrative suggests that influence is computed on a per-example basis to adjust the attention dynamically (delta). Yet, the inclusion of plots and tables depicting a consistent delta value across examples implies that this parameter could potentially be determined via straightforward heuristics or a hyper-parameter search. This raises questions about the essentiality of influence in the proposed method. If influence is not critical to the functionality, a direct comparison of influence and traditional attention rollout—perhaps in basic settings like verb singularity classification—would be imperative to justify its mention as a key contribution. In summary, my recommended experiments would be:
- Add another line for the experiments with "flexible delta based on influence"
- Comparing the performance of influence and attention rollout on the task of localizing the attributed attention for singular verb classification with the same setting than the original attention rollout paper.

Sec 4:
1) The experiments seem to employ a limited set of prompt templates, which might not fully capture the model's versatility in handling varied instructions. The effect of different prompt structures and the positioning of instructions within sentences could potentially alter the outcomes, suggesting a need for a broader range of templates in evaluation.

2) The choice of p(output in French) as the only metric, while straightforward, does not address the quality of the summarization. Considering the discussion on potential issues with high attention augmentation (e.g., delta=2 leading to nonsensical outputs), it is vital to also assess the quality of the generated summaries, either through human evaluations or automated metrics, to ensure the practical utility of the model outputs.

3) holistic evaluation of the impact: similar to weakness 4.2, there is a need for a more comprehensive evaluation of how attention augmentation influences the model’s overall comprehension and performance. Specifically, how does attention augmentation affect the comprehension of the other part of the input? Does it degrade the general performance? Could it be used my malicious users to hack and jailbreak the aligned LLM by forcing attention at unintended tokens? I understand that these evaluations may require a lot of effort but I believe they are crucial questions to be answered to claim the method to be truly useful. Here are some proposed experiments that you may try:
- comprehension: Add a dataset that requires complex reasoning to see if paying heavy attention to the instruction would break the performance.
- safety: Pick an aligned model and test it on an adversarial attack benchmark. Two ways of attack by hacking the attention augmentation: 1. Wrapping the adversarial instruction with the attention special token. 2. Wrapping a random 2/3 of the tokens in the input to significant increases the temperature.

**Questions:**

Here are some questions that I think will clarify the purpose of the proposed method:

1) Does SFT achieve similar effect on the attention distribution (I suspect not) than directly augmenting them? If not, you can further demonstrate the superiority of the method as it may generalize on new instruction much better than SFT.
2) Details of needle in a haystack experiment is a bit ambiguous. Is the "attention token" added across the needle token or the instruction tokens? If it's the needle token, it does not make much sense to demonstrate this as an augmentation to the instruction-following skills as you are essentially marking out the key information to the answer here.
3) What does the y-axis label (Depth) of Fig 4 in section 4 mean? Could you provide a clearer definition in the section about what is 'Depth' and how is that related to the task of 'needle in a haystack'?

Here are some questions for the future exploration:

1) Solution to lost in the middle: with something like a normal distribution of delta, could you use attention augmentation as a general solution to "lost in the middle" for long context inference?
2) For LLMs trained with structured prompt that highlights the instruction tokens in the input (Llama3-instruct for instance), would the method still work?

I am happy to adjust my evaluation if the questions and weaknesses are discussed. I think it is an interesting paper that just needs to be polished a bit more.

---

### Note · Authors · 2024-11-15

**Comment:**

We appreciate the insightful feedback and valuable suggestions provided by the reviewers and Area Chairs. After thorough reflection, we have chosen to withdraw our submission to dedicate time to further enhancing and refining our paper in light of these constructive critiques. Thank you once more for your thoughtful evaluation and the effort you invested in reviewing our work.

**Withdrawal Confirmation:**

I have read and agree with the venue's withdrawal policy on behalf of myself and my co-authors.